# Heat Stress Impact on Yield and Composition of Quinoa Straw under Mediterranean Field Conditions

**DOI:** 10.3390/plants10050955

**Published:** 2021-05-11

**Authors:** Javier Matías, Verónica Cruz, María Reguera

**Affiliations:** 1Agrarian Research Institute “La Orden-Valdesequera” of Extremadura (CICYTEX), 06187 Badajoz, Spain; veronica.cruz@juntaex.es; 2Department of Biology, Universidad Autónoma de Madrid, c/Darwin 2, Campus de Cantoblanco, 28049 Madrid, Spain; maria.reguera@uam.es

**Keywords:** quinoa, stems, high temperatures, food security, climate smart agriculture, quinoa by-products

## Abstract

Quinoa (*Chenopodium quinoa* Willd.) is receiving increasing attention globally due to the high nutritional value of its seeds, and the ability of this crop to cope with stress. In the current climate change scenario, valorization of crop byproducts is required to support a climate-smart agriculture. Furthermore, research works characterizing and evaluating quinoa stems and their putative uses are scarce. In this work, straw yield and composition, and the relative feed value of five quinoa varieties, were analyzed in two consecutive years (2017–2018) under field conditions in Southwestern Europe. High temperatures were recorded during the 2017 growing season resulting in significantly decreased straw yield and improved feed value, associated with compositional changes under elevated temperatures. Crude protein, ash, phosphorus, and calcium contents were higher under high temperatures, whereas fiber contents decreased. The relative feed value was also higher in 2017 and differed among varieties. Differences among varieties were also found in straw yield, and contents of phosphorus, potassium, and calcium. Overall, the results presented here support a sustainable quinoa productive system by encouraging straw valorization and shedding light on the mechanisms underlying heat-stress responses in this crop.

## 1. Introduction

In the twenty-first century, climate change has become one of the most critical global challenges, and is particularly crucial for the agricultural and livestock sectors [1,2,3,4]. This phenomenon is associated with increments of atmospheric greenhouse gas (GHG) emissions and the increase in the mean global temperature, in addition to alterations in the precipitation regime, affecting food and feed production, and its nutritional composition [1,3,5,6,7,8]. Because these changes represent a serious threat to global food security, it is necessary to understand how crops respond to elevated temperatures and how their tolerance to heat stress can be improved [9]. In particular, the Mediterranean climate comprises the western edges of continents between latitudes of 30° and 45° [10,11,12]. Regions with a Mediterranean climate are characterized by hot and dry summers, mild wet winters, and low and irregular rainfall. Because of climate change, severe droughts, increased temperatures, and salinity problems are predicted to be more frequent in the near future in these areas [13].

In addition, the current rapid growth in the global population is expected to lead to a sharp increase in demand for food in coming decades [14]. Furthermore, available land for agriculture and water resources is increasingly scarce following soil degradation and other land uses [4,15]. Hence, agriculture in the twenty-first century must meet increasing food demand with fewer resources, while GHG emissions should be reduced. Within this context, it is necessary to introduce crops with lower irrigation requirements (e.g., in South Europe alone, more than 60% of freshwater is used by agriculture [13]) and better adapted to less favorable soil conditions, in addition to valorizing crop byproducts. The latter can be achieved if these materials are first well characterized, shedding light on the effects of climate change on the nutritional value of plants. 

Quinoa (*Chenopodium quinoa* Willd.) is an ancient crop from South America that has expanded globally because of the increasing interest in the nutritional composition of its seeds. This has contributed to its selection as one of the crops destined to contribute to food security in the next century, according to the FAO (Food and Agricultural Organization) [16]. Quinoa has also been studied for its potential use as a forage crop because of the high nutritional value of the whole plant for livestock [17]. However, this use has been little explored. In contrast, quinoa seeds are well known for being gluten-free, because of their high-quality protein and because they are one of the few plant foods that contain all nine essential amino acids, with a wider amino acid profile than cereals or legumes [18]. Quinoa is an annual C3 plant belonging to the Amaranthaceae family [19], with remarkable adaptability to unfavorable growing conditions [20]. It is a facultative halophytic plant species that is also tolerant to the combined effect of high temperatures and salinity [21,22,23], and is considered an environmental stress-resilient crop plant [17].

Quinoa has been cultivated for thousands of years in the Andean region comprising Peru, Bolivia, Colombia, and Ecuador although currently its cultivation has expanded globally [6,24]. Indeed, there is a remarkable interest in growing quinoa at European latitudes [25]. In the Mediterranean region, quinoa has been cultivated in Spain, Italy, Greece, and Portugal. In Spain, quinoa cultivation has been expanded significantly in recent years, particularly in the southern part of the country. Interestingly, one of the consequences of the massive expansion of quinoa cultivation is the generation of residues composed of quinoa plant straw, for which possible uses have been little explored [15,26]. Only recently has the use of quinoa straw been evaluated for animal feed [15,17] or generation of renewable energy [15,26].

Thus, in this study, aiming to further explore possible uses of quinoa crop byproducts, and therefore contribute to the implementation of sustainable quinoa cultivation under Mediterranean conditions, the straw production and composition of five quinoa varieties were evaluated in a field located in Southwestern Spain, where episodes of elevated temperatures are frequent. Differences in straw biomass and composition were expected to occur based on the developmental differences of the genotypes tested. Furthermore, the impact of high temperatures under field conditions was analyzed. These results will contribute to a better understanding of how temperature stress might influence straw production.

## 2. Results

### 2.1. Straw Yield and Biomass Partitioning

As observed in Figure 1, the straw yield varied significantly according to the year and variety. Interaction between both factors was also significant. The mean straw yield was significantly higher in 2018 (2.2 t ha^−1^) than that in 2017 (1.7 t ha^−1^), except for Roja and Duquesa, which achieved lower yields in 2018. Marisma reached the highest mean stem yield among varieties (2.8 t ha^−1^), and was remarkably high in 2018 (3.9 t ha^−1^). Biomass partitioning was evaluated by determining the HI index, which was significantly influenced by the variety and the year (Figure 2). The average HI in 2017 (0.49) was lower than that in 2018 (0.53), mainly due to the 20% HI reduction achieved in 2017 by Roja (0.40) and Duquesa (0.45). The highest average HI (0.55) was achieved by Jessie (short cycle), whereas the lowest HI was obtained by the varieties with longer life cycles (Roja: 0.48 and Duquesa: 0.50). HI was not correlated with straw yield. 

### 2.2. Crude Protein (CP), Crude Fibre (CF), and Ash Content 

The CP and CF contents were significantly influenced by the year, whereas no significant differences were found according to the variety (Table 1). The CP content in 2017 (12.8%) was 52.3% higher than that in 2018 (8.4%). On the contrary, the CF content was significantly higher in 2018 (33.8%) compared to 2017 (28.0%). The average ash content differed significantly according to the variety. The average ash content varied from 11.5% (Roja) to 16.6% (Jessie) with no different found between years.

### 2.3. Fibre Composition 

Significant higher neutral detergent fiber (NDF), acid detergent fiber (ADF), and acid detergent lignin (ADL) contents were determined in 2018 (55.4%; 40.5%; 6.5%, respectively) compared to 2017 (44.1%; 30.8%; 5.2%, respectively), as shown in Table 2. The variety showed a significant influence on the ADF and the ADL content. The average ADF content ranged from 31.6% (Jessie) to 40.9% (Roja), whereas the average ADL content varied from 5.3% (Marisma) to 6.5% (Roja). Interactions between the year and variety were not significant. The hemicellulose (HEM) contents only showed significant differences in 2017 according to the variety, ranging from 11.2% (Marisma) to 15.2% (Jessie). Significant differences were also found in cellulose (CEL) contents in 2017 according to the variety, ranging from 20.6% (Jessie) to 29.2% (Pasto). The mean CEL content was significantly affected by the year and the variety. It was higher in 2018 (34.0%) than in 2017 (25.6%), and ranged from 25.3% in Jessie to 34.5% in Roja. Neither the HEM nor the CEL contents were significantly affected by the year × variety interaction.

### 2.4. Mineral Composition

As can be observed in Table 3, the N, P, and Ca contents differed significantly according to year, and all of them were higher in 2017. The variety had a significant influence on the P, K, and Ca contents, but not on the N and Mg average contents, which were 1.7% and 0.64%, respectively. The average P content ranged from 0.17% (Roja and Duquesa) to 0.24% (Jessie). K showed average contents from 4.4% (Roja) to 6.0% (Jessie), and the average of Ca content varied between 0.8% (Roja) and 1.6% (Pasto).

Interactions between year and variety were significant for the P and Ca contents. In addition, the P content was relatively similar in both years in the short cycle variety (Jessie), whereas in the other varieties, the P content was considerably higher in 2018, especially in Roja (medium-long cycle), for which differences were found to be higher than 60%. Regarding the Ca content, differences between years were small in Jessie and Pasto, whereas in 2018 the Ca content decreased by more than 18% in the other varieties.

### 2.5. Relative Feed Value

The year showed a significant influence on the digestible dry matter (DDM), the dry matter intake (DMI) and on the relative feed value (RFV), which were higher in 2017 (Table 4). The variety significantly influenced the DMI and RFV values, achieving the highest values in the short cycle variety Jessie (64.3 and 131.0, respectively), and the lowest values in the medium-long cycle varieties (Roja; 57.1 and 99.4, respectively). Interaction between the year and the variety was not significant.

## 3. Discussion

The use of quinoa straw has the potential to contribute to the development of a cleaner agriculture. This byproduct has been used and studied for agricultural purposes as feed and bedding material, in addition to bioenergy or biomaterial production [27,28]. Quinoa cultivation has experienced a rapid increase in the past decade, in parallel with its expansion to many different geographical areas around the world [29]. However, quinoa straw use has been little explored because the interest in this crop has focused mainly on seed yield. Thus, the current study aimed to evaluate variations in quinoa straw characteristics linked to genetic differences under Mediterranean field conditions, to implement alternative uses of this crop and therefore contribute to the development of sustainable agriculture. This study was performed in two consecutive years showing different climatic conditions, which included average differences in Tmax during the quinoa growth period of 5 °C. Generally, elevated temperatures inhibit quinoa plant growth by impacting quinoa flowering [19,30,31,32]. Heat stress can limit the source and sink capacity of plants, reducing growth and development [33]. Indeed, the higher temperatures registered during the first year resulted in a reduction of straw yield in 2017 (1.7 t ha^−1^) compared to 2018 (2.1 t ha^−1^) (Figure 1), especially in short-medium cycle varieties, such as Marisma, which doubled its yield in 2018. On the contrary, long-cycle varieties, such as Roja or Duquesa, showed lower straw yields in 2018, which may point to a differential response depending on the developmental stage affected by high temperatures. In cereals, such as wheat, it has been reported that heat stress is an important factor that reduces straw yield, because photosynthesis is altered [34,35]. Nonetheless, further analysis should be performed to determine the exact impact of heat stress on the vegetative growth and its influence on straw yield, particularly in quinoa. It should be noted that the high straw yield achieved by Marisma (2.8 t ha^−1^, on average), especially in 2018 (3.9 t ha^−1^), was well correlated with a higher seed yield, as previously reported [36], which is interesting from an agronomical perspective because seed yield penalties should be avoided. Intriguingly, the two-year average straw yield (1.9 t ha^−1^) achieved in this work was lower than those previously reported for Mediterranean field conditions (3.0 t ha^−1^ for Titicaca; 7.4 t ha^−1^ for Regalona) [37], and similar to those obtained by Asher et al. (2020), although at the bottom of the range (0.5–9.1 t ha^−1^). The results reported earlier can be partially explained by the lower HI obtained in these works, which resulted in higher straw yield per seed yield. This was probably a consequence of the different environmental conditions, and differences in the planting density and/or in the genotypes used, that resulted in detrimental effects on seed yield but positive effects on vegetative growth [38,39]. In the current study, the higher temperatures registered in 2017 also had a significant impact on the HI in medium-long cycle varieties (Roja and Duquesa). Due to the longer vegetative stage of Roja and Duquesa, flowering coincided with higher Tmax and lower RH, increasing the probability of flower damage causing a HI decrease (Figure 2). On the contrary, the shortest-cycle genotype Jessie showed the highest HI (0.55) in 2017, which was related to its ability to escape from high temperatures at flowering stage, which resulted in the maintenance of its sink capacity but larger straw yield penalties. In general, as shown in Figure 2, the average HI was 0.51, which is similar to the HI reported in modern wheat varieties (0.3–0.6) [39,40], and higher than those reported for grain legumes and canola [41]. Nonetheless, no correlation between straw yield and HI was found, in contrast to the findings of previous studies [42,43].

The straw composition differed between years as observed in Table 1, Table 2, Table 3 and Table 4, probably influenced by the high temperatures of 2017. This effect has been previously studied in the main products of different crops [44,45,46,47,48], although the impact on byproducts, such as straw, has been little explored. In fact, the current study is pioneering in its evaluation of the effect of high temperatures on quinoa straw yield and composition. Furthermore, the CP content range in the straws obtained in this study (6.9–13.5%) was slightly higher than that obtained by Asher et al. (2020) under Mediterranean conditions (5.1–10.6%).

As can be observed in Table 1, the CP content was considerably higher in 2017 (52.7%). Because of the higher average straw yield of 2018, the lower straw CP content achieved in 2018 could have been a consequence of a dilution effect. However, in 2017 the CP content was also higher in the medium-long cycle varieties, which also achieved higher straw yields. Therefore, high temperatures could lead to higher straw CP content independently of the straw yield. Interestingly, no correlation was found between CP content and the other parameters analyzed in this study, except for the straw N content. It should be noted that the high CP content observed here (particularly in 2017) was also higher than the contents found in cereal (2.9%) or legume (7.4%) straws [49], which is important when the aim is to use this byproduct for animal feed. In contrast, the CF content was reduced about 20% in 2017 without a significant effect linked to the variety (Table 1). Plant fiber includes the cell wall used for providing mechanical support to the plant, and the vascular system in which fluids are transported [50,51]. The response observed in the CF content was likely related to a change in composition due to the heat stress impact on the biosynthesis-related pathways of cell wall components [52,53,54]. Actually, a steeper decrease was detected in the ADF content (with an average reduction of 24%), and in the cellulose content (which showed an average decrease of 25%) (Table 2). The two-year average of NDF was slightly lower than the values determined by Asher et al. 2020 (41.4–63.6% and 44.4–71.3%, respectively), whereas the lignin was higher than the values reported by these authors (5.3–7.0% and 5.03–7.83%, respectively). This effect raises an interesting aspect related to the straw composition as lignocellulosic biomass, which is now considered an important fiber resource for renewable energy and biomaterial production [55]. By comparison, the ash content did not change between years, achieving values close to 15%, restricting quinoa straw’s use as a solid biofuel (for which ash should not exceed 1.5%) [56].

When analyzing the increased mineral and protein contents in 2017, the changes could point to a detrimental effect of higher temperatures in nutrient translocation into the seed as observed in wheat [33]. However, the protein and mineral contents were also higher in quinoa seeds in 2017 [36]. It is known that the environmental conditions can affect nutrient levels of quinoa seeds [57]. In the current study, the straw N and P contents were significantly lower in 2018, which could be related to a dilution effect linked to higher yields, similar to that described in cereal grains [58]. However, those differences were even higher in Roja and Duquesa, which achieved higher yields in 2017. Furthermore, the straw Ca content was significantly lower in 2017, especially in Roja and Duquesa. Therefore, differences in mineral nutrient contents were probably more related to a still unknown heat-induced adaptation mechanism (and/or to the effect caused by the interaction between nutrients) than to a direct effect caused by nutrient dilution. The effect of heat stress on cellular osmotic adjustments due to the increased transpiration rates when elevated temperatures occur should also not be discounted; this could result in increments in the mineral and protein content [59,60]. Interestingly, the mineral composition of quinoa straw was richer from a nutritional perspective compared with the mineral composition of cereal straws commonly used for animal feed [61]. For instance, Ca or Mg contents in wheat straws were reported to be, on average, 0.18% and 0.06%, respectively whereas quinoa contents were, on average, 1.2% and 0.64%, respectively (Table 3).

The relative feed value (RFV) developed by the Universities of Minnesota and Wisconsin and the American Forage and Grassland Council (AFGC), is an index widely used to determine the forage quality, which combines significant nutritional factors (including voluntary intake and digestibility) [62,63]. When evaluating the changes in composition of the quinoa straw, a significant impact on the RFV was detected, which resulted in the increase in this index in 2017, due to the lower NDF and ADF, improving the straw nutrient intake and digestibility (Table 4). Considering the AFGC classification method [62], the mean value for the RFV in 2017 (138.5) would indicate that this byproduct would be classified as a forage type I (125–151), which is considered acceptable. In 2018, the mean value (98.5) would correspond to a low-quality forage (being classified as a type IV forage (87–102). Therefore, based on these results, in addition to the higher CP and mineral contents (higher than those found in winter cereals, which is the straw most frequently used in animal feed), it can be considered that the quinoa straw obtained in 2017 possesses a higher nutritional value. Nonetheless, further research is required to determine the exact impact of heat stress on straw yield and composition. Based on the results presented here, elevated temperatures, which are expected to continue increasing due to the continuation of the warming period [64], may positively impact straw composition in a genotype-dependent manner.

Overall, the results here presented show that Marisma is the variety with the best cultivation potential in terms of straw yield for this particular area (Southwestern Spain), despite suffering important yield penalties linked to the elevated temperatures suffered in 2017 [36]. Furthermore, the straw composition analysis performed shows that quinoa straw is a valuable resource for animal feed, biofuel, or biomaterial production. To enhance quinoa straw valorization, it is crucial to evaluate the straw yield and compositional changes related to the genotype and environment, and to understand the relationships among biomass, seed, and straw yield. Our results, together with the previously reported changes in chemical composition of cereal straws, highlight that quinoa straw yield and composition may be affected by agronomic and genotypic factors, and environmental and climate conditions [50,55]. Within the current climate context in which more frequent episodes of elevated temperatures in the Mediterranean area are expected to occur, and considering that the effects of heat stress on plants trigger complex responses that result in the alteration of growth and development, thereby changing physiological functions and reducing seed/grain formation and plant yield [33,65], further studies should be undertaken to evaluate the impact of heat stress on straw yield and composition, with the aim of selecting the best adapted quinoa cultivars for a particular area of cultivation. More importantly, this work points to the many possibilities offered by the use of quinoa straw as an agricultural byproduct that can greatly contribute to sustainable agriculture.

## 4. Materials and Methods

### 4.1. Location, Climate and Soil Characteristics of the Experimental Site

A two-year field experiment was conducted during 2017–2018 at the experimental farm of the Center for Scientific and Technological Research of Extremadura (CICYTEX), located in Southwest Spain (lat. 38°51′10″ N; long. 6°39′10″ W). Data of monthly mean minimum and maximum temperature (Tmin and Tmax), and the rainfall during the crop cycle (Appendix A) were obtained from the weather station located at the experimental farm. The soil was a sandy loam, neutral (pH 6.9), presenting 0.38% organic matter, 0.24% total N, and 93.4 ppm and 57.9 ppm of available P and K, respectively.

### 4.2. Layouts of Experiments, Plant Material and Crop Management

Five European varieties of quinoa (Pasto, Marisma, Jessie, Roja, Duquesa) were evaluated in a randomized complete block design with four replications: Jessie (short cycle; ~120 d); Pasto and Marisma (medium cycle; ~135 d); Roja and Duquesa (medium-long cycle; ~145 d). The plot size was formed by four rows 0.75 m apart and 10 m long. Sowing was conducted mechanically in early February, at a dose of 4 kg ha^−1^. Weeding was carried out by hand when required. Irrigation was carried out by sprinkling to maintain the soil under non-limiting water conditions. The crop was fertilized at the rate of 150, 100, and 100 kg/ha of N, P_2_O_5_, K_2_O, respectively. Plants were harvested by hand at physiological maturity (in the middle of June for Jessie, and early July for the other four varieties). The sampling area was 3 m^2^. A stationary thresher was used to collect the seed. The straw samples were ground through a 1 mm screen for further analysis.

### 4.3. Analysis and Measurements

Data were expressed on a dried weight (dW) basis. Analysis of moisture, crude protein (CP), and ash contents were conducted following the AOAC Official Methods [66]. The mineral content was assessed following the official methods of the Spanish Ministry of Agriculture [67]. P was analyzed by a UV-VIS spectrophotometer (Hitachi U-2810). K was determined using flame atomic emission spectroscopy, and Ca and Mg by flame atomic absorption (SpectrAA 110, Agilent). The crude fiber (CF), neutral detergent fiber (NDF), acid detergent fiber (ADF), and acid detergent lignin (ADL) values were analyzed following the Ankom procedure (ANKOM Technology, Fairport, NY, USA), using a fiber analyzer (ANKOM 2000) and F57 Ankom filter bags (porosity: 25 µm). Hemicellulose (Hem) and cellulose (Cel) contents were calculated as follows:$$\mathrm{Hemicellulose}\;\left(\mathrm{Hem}\right)\;=\;\left[\mathrm{NDF}\%\;-\;\mathrm{ADF}\%\right]$$
$$\mathrm{Cellulose}\;\left(\mathrm{Cel}\right)\;=\;\left[\mathrm{ADF}\%\;-\;\mathrm{ADL}\%\right]$$

Relative feed value (RFV) was calculated as calculated from the digestible dry matter (DDM) and the dry matter intake (DMI) (live weight: LW, %) according to the following equations:$$\mathrm{DDM}\;\left(\%\right)\;=\;88.9-\left[0.779\;\times \;\mathrm{ADF}\%\right]\;$$
$$\mathrm{DMI}\;\left(\mathrm{Live}\;\mathrm{Weight}:\;\mathrm{LW}\;\%\right)\;=\;120/\left[\mathrm{NDF}\%\right]\;$$
$$\mathrm{RFV}\;=\;\left[\mathrm{DMD}\;\times \;\mathrm{DMI}\right]/1.29$$

Harvest index (HI) was calculated as the ratio between the grain yield (G) and the yields of straw plus grain (G + S), in order to determine the biomass partitioning index, as previously reported [36].

### 4.4. Statistical Analysis

All measured and derived data were processed using a two-way analysis of variance (ANOVA), including the year, the variety, and their interactions in the model. The year was treated as a fixed factor. Normality and equal variances could be assumed, according to the results of the Shapiro Wilk test and Levene’s test, respectively. For better interpretation of the data, additional one-way ANOVA was carried out for each year separately. When the F ratio was significant (*p* < 0.05), Tukey’s test was performed and used to compare means. Student’s *t*-test was used to determine statistically significant differences between years for each variety. Analyses were performed using the Statistix 8 analytical software.

## Figures and Tables

**Figure 1 plants-10-00955-f001:**
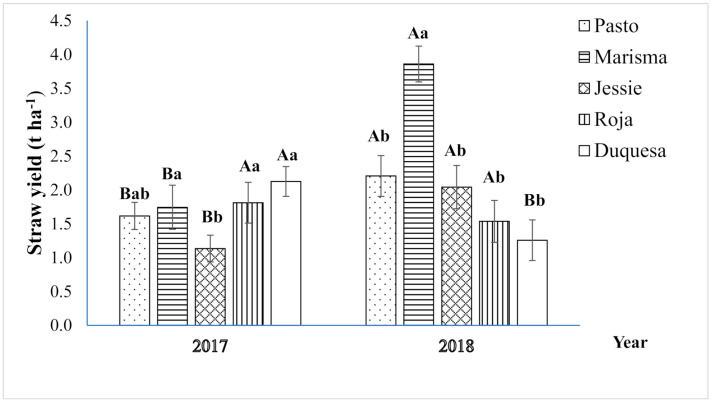
Straw yield (t. ha^−1^) of five quinoa varieties during the two years (2017 and 2018) of field experiments. Error bars represents the standard deviation. Different uppercase letters in the same variety indicate a significant difference between years according to Student’s *t*-test at *p <* 0.05. In each year, different lowercase letters indicate significant differences among varieties according to Tukey’s test at *p <* 0.05.

**Figure 2 plants-10-00955-f002:**
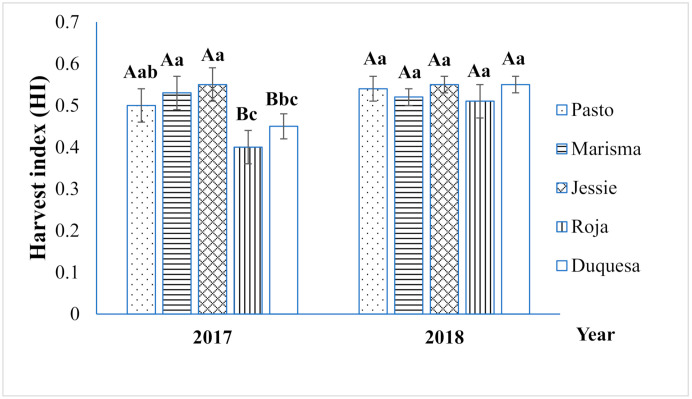
Harvest index (HI) of five quinoa varieties during the two years (2017 and 2018) of field experiments. Error bars represent the standard deviation (SD). Different uppercase letters in the same variety indicate a significant difference between years according to Student’s *t*-test at *p <* 0.05. In each year, different lowercase letters indicate significant differences among varieties according to Tukey’s test at *p <* 0.05.

**Table 1 plants-10-00955-t001:** Crude protein (CP), crude fiber (CF), and ash content (%) of straw of five quinoa varieties during the two years (2017 and 2018) of field experiments.

Variety	CP^1^ (%)			CF^2^ (%)			Ash (%)		
	2017	2018	Mean	2017	2018	Mean	2017	2018	Mean
Pasto	11.7	7.6	9.7	29.8	32.5	31.1	15.1	15.7 ab	15.4 abc
Marisma	13.5	8.1	10.8	27.8	32.3	30.0	16.7	14.6 ab	15.6 ab
Jessie	12.7	10.5	11.6	24.7	30.1	27.4	17.4	15.9 a	16.6 a
Roja	13.2	6.9	10.0	30.3	39.2	34.7	13.0	12.6 b	12.8 c
Duquesa	13.1	8.4	10.8	27.6	35.0	31.3	14.4	13.0 ab	13.7 bc
Mean	12.8 A	8.4 B	10.6	28.0 B	33.8 A	30.9	15.3	14.3	14.8
HSD	4.0	5.3	4.4	6.5	16.2	7.8	4.6	3.2	2.6
**Significance**									
Year (Y)			**			*			n.s.
Variety (V)	n.s.	n.s.	n.s.	n.s.	n.s.	n.s.	n.s.	*	**
Y × V			n.s.			n.s.			n.s.

CP^1^: crude protein; CF^2^: crude fiber. Variety means denoted by different lowercase letters in the same column are significantly different at *p* < 0.05 according to Tukey’s test. Year means followed by different uppercase letters are significantly different at *p* < 0.05 according to Tukey’s test. HSD: critical value for comparison. n.s.: not significant; significant at * *p* < 0.05; ** *p* < 0.01, respectively.

**Table 2 plants-10-00955-t002:** Fiber composition of straw of five quinoa varieties during the two years (2017 and 2018) of field experiments.

Variety	NDF^1^ (%)			ADF^2^ (%)			ADL^3^ (%)			HEM^4^ (%)			CEL^5^ (%)		
	2017	2018	Mean	2017	2018	Mean	2017	2018	Mean	2017	2018	Mean	2017	2018	Mean
Pasto	47.2	52.2	49.7	34.2	38.0	36.1 ab	5.1 ab	6.0	5.5 ab	13.0 ab	14.2	13.6	29.2 a	32.0	30.6 ab
Marisma	41.4	53.7	46.7	30.3	37.9	34.1 ab	4.5 b	6.1	5.3 b	11.2 b	15.8	13.5	25.8 ab	31.8	28.8 ab
Jessie	41.8	51.7	47.6	26.6	36.6	31.6 b	6.0 a	6.6	6.3 a	15.2 a	15.0	15.1	20.6 b	30.0	25.3 b
Roja	48.3	63.6	46.7	34.0	47.8	40.9 a	5.9 a	7.0	6.5 a	14.4 a	15.8	15.1	28.1 ab	40.8	34.5 a
Duquesa	42.1	55.7	48.9	28.8	42.1	35.5 ab	4.6 ab	7.0	5.8 ab	13.3 ab	13.6	13.5	24.2 ab	35.1	29.7 ab
Mean	44.1 B	55.4 A	49.8	30.8 B	40.5 A	35.6	5.2 B	6.5 A	5.9	13.4	14.9	14.2	25.6 B	34.0 A	29.8
HSD	11.4	19.5	10.0	9.3	16.2	8.3	1.4	1.4	0.9	3.1	4.2	2.5	8.3	15.1	7.7
**Significance**															
Year (Y)			**			**			**			n.s.			**
Variety (V)	n.s.	n.s.	n.s.	n.s.	n.s.	*	*	n.s.	**	*	n.s.	n.s.	*	n.s.	*
Y × V			n.s.			n.s.			n.s.			n.s.			n.s.

NDF^1^: neutral detergent fiber, ADF^2^: acid detergent fiber, ADL^3^: acid detergent lignin, HEM^4^: hemicellulose, CEL^5^: cellulose. Variety means denoted by different lowercase letters in the same column are significantly different at *p* < 0.05 according to Tukey’s test. Year means followed by different uppercase letters are significantly different at *p* < 0.05 according to Tukey’s test. HSD: critical value for comparison. n.s.: not significant; significant at * *p* < 0.05; ** *p* < 0.01, respectively.

**Table 3 plants-10-00955-t003:** Mineral composition of straw of five quinoa varieties during the two years (2017 and 2018) of field experiments.

Variety	N (%)			P (%)			K (%)			Ca (%)			Mg (%)		
	2017	2018	Mean	2017	2018	Mean	2017	2018	Mean	2017	2018	Mean	2017	2018	Mean
Pasto	1.9	1.2	1.5	0.23	0.15 b	0.19 ab	4.3 b	5.1	4.7 ab	1.6 a	1.5 a	1.6 a	0.62	0.68	0.65
Marisma	2.2	1.3	1.7	0.28	0.15 b	0.22 ab	4.6 ab	5.0	4.8 ab	1.8 a	1.1 abc	1.5 ab	0.82	0.69	0.76
Jessie	2.0	1.7	1.9	0.23	0.25 a	0.24 a	6.1 a	5.8	6.0 a	1.1 b	1.2 ab	1.2 bc	0.63	0.77	0.70
Roja	2.1	1.2	1.6	0.25	0.09 b	0.17 b	4.6 ab	4.1	4.4 b	0.9 b	0.6 c	0.8 d	0.60	0.45	0.53
Duquesa	2.1	1.3	1.7	0.22	0.12 b	0.17 b	5.1 ab	4.9	5.0 ab	1.1 b	0.9 bc	1.0 cd	0.63	0.55	0.59
Mean	2.0 A	1.3 B	1.7	0.24 A	0.15 B	0.20	5.0	5.0	5.0	1.3 A	1.1 B	1.2	0.66	0.63	0.64
HSD	0.6	1.0	0.5	0.10	0.09	0.06	1.7	2.6	1.4	0.4	0.5	0.3	0.38	0.34	0.26
**Significance**														
Year (Y)			**			***			n.s.			*			n.s.
Variety (V)	n.s.	n.s.	n.s.	n.s.	**	*	*	n.s.	*	***	**	***	n.s.	n.s.	n.s.
Y × V			n.s.			**			n.s.			**			n.s.

Variety means denoted by different lowercase letters in the same column are significantly different at *p* < 0.05 according to Tukey’s test. Year means followed by different uppercase letters are significantly different at *p* < 0.05 according to Tukey’s test. HSD: critical value for comparison. n.s.: not significant; significant at * *p* < 0.05; ** *p* < 0.01 and *** *p* < 0.001, respectively.

**Table 4 plants-10-00955-t004:** Straw forage quality of five quinoa varieties during the two years (2017 and 2018) of field experiments.

Variety	DDM^1^			DMI^2^			RFV^3^		
	2017	2018	Mean	2017	2018	Mean	2017	2018	Mean
Pasto	2.6	2.3	2.4	62.3	59.3	60.8 ab	124.2	106.3	115.2 ab
Marisma	2.9	2.3	2.6	65.3	59.4	62.3 ab	147.5	104.1	125.8 ab
Jessie	2.9	2.3	2.6	68.2	60.3	64.3 a	152.7	109.3	131.0 a
Roja	2.5	1.9	2.2	62.4	51.7	57.1 b	120.9	77.9	99.4 b
Duquesa	2.9	2.2	2.5	66.5	56.1	61.3 ab	147.1	94.9	121.0 ab
Mean	2.7 A	2.2 B	2.5	64.9 A	57.4 B	61.1	138.5 A	98.5 B	118.5
HSD	0.7	0.7	0.5	7.2	12.7	6.5	49.6	51.4	31.5
**Significance**									
Year (Y)			**			**			**
Variety (V)	n.s.	n.s.	n.s.	n.s.	n.s.	*	n.s.	n.s.	*
Y × V			n.s.			n.s.			n.s.

DDM^1^: digestible dry matter; DMI^2^: dry matter intake; RFV^3^: relative feed value. Variety means denoted by different lowercase letters in the same column are significantly different at *p* < 0.05 according to Tukey’s test. Year means followed by different uppercase letters are significantly different at *p* < 0.05 according to Tukey’s test. HSD: critical value for comparison. n.s.: not significant; significant at * *p* < 0.05; ** *p* < 0.01, respectively.

## Data Availability

The data that support the findings of this study are available upon request.

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
