# Peer review of "Heat Stress Impact on Yield and Composition of Quinoa Straw under Mediterranean Field Conditions"

_plants, 2021, doi:10.3390/plants10050955_

Round 1
Reviewer 1 Report
The paper reports the quality of quinoa straw and its response to two years with different climate.
This is a relevant area of research since the use of agricultural by-products is a goal to achieve sustainability for the future and is within the scope of Plants; however, it shows some concerns.
The article is too descriptive, limiting to highlight the results and their importance, instead of trying to analyze how they happened.
In the tables there is no standard deviation or standard error, so the accuracy of the data is hard to interpret. Additionally, no comment is offered as to whether the authors checked the assumptions of the anovas.
Minor comments:
Line 47-48: The parenthesis opens, but does not close
Line 234: “the lower straw CP achieved in 2017 could…” It should be: “the lower straw CP achieved in 2018 could…”
Line 358-359: The Harvest index should be better explained.
Author Response
Many thanks to Reviewer 1. The comments made have greatly helped improving this manuscript.
Changes in the manuscript have been highlighted by yellow color. The overall language of the manuscript has been revised, and changes have been highlighted by text in a red colour font.
Response to Reviewer 1
The article is too descriptive, limiting to highlight the results and their importance, instead of trying to analyze how they happened.
After revising the manuscript, we believe that the analysis of the presented data has improved. We hope this has helped to analyze the processes and responses that occurred.
In the tables there is no standard deviation or standard error, so the accuracy of the data is hard to interpret.
Thanks Reviewer 1. Standard deviation (or error) was not included to make easier the comprehension of the data presented since there are multiple parameters and letters added to show significant differences. Nonetheless, for each parameter, the Tukey HSD was added, which is directly correlated with the error. This reflects, indirectly, the variability of the data allowing the interpretation of the results.
Additionally, no comment is offered as to whether the authors checked the assumptions of the anovas.
Reviewer 1 is right. The comment has been included in lines 366-367: Normality and equal variances could be assumed, according to the results of Shapiro Wilk test and a Levene’s test, respectively
Minor comments:
Line 47-48: The parenthesis opens, but does not close
Thanks. This has been corrected
Line 234: “the lower straw CP achieved in 2017 could…” It should be: “the lower straw CP achieved in 2018 could…”
Thanks. This has been corrected
Line 358-359: The Harvest index should be better explained.
Thanks Reviewer 1, we have modified the sentence accordingly (Please, check lines 359-360).
Reviewer 2 Report
Heat stress impact on yield and composition of quinoa straw under Mediterranean field conditions
The investigation is analyzing yield and straw composition of five European quinoa varieties. The manuscript is aimed to explore future possible uses of quinoa crop byproducts under Mediterranean conditions giving the new insights into subject, therefore the investigation is interesting. The authors used a great number of methods to determine straw yield, protein, fibre and ash content, fibre and mineral composition as well as relative feed value in order to provide valuable information for animal feeding and biomaterial production. However, there are some questions that authors have to describe in more detail.
- The hypothesis is not clearly stated, and it should be set out and elaborated. Also, authors should point out aims in more details.
- Line 47: there is bracket that is either unnecessary or it should be closed
- It is unusual to use letters to describe statistical difference in t-test, are the authors using letters to be easier to read the statistics and to distinguish differences between each variety in 2017 and 2018?
- Why the authors calculate mean values for each parameter for both year and for all varieties in single year? Please explain since I don`t see the point of that.
- Line 290: reference 62 should be corrected
- Methods for analysis and measurements should be described more and references should be added.
Best regards
Author Response
Many thanks to Reviewer 2. We hope that, after revising and respond to her/his comments, we have clarified the data and analysis presented.
Changes in the manuscript have been highlighted by yellow color. The overall language of the manuscript has been revised, and changes have been highlighted by text in a red colour font.
Response to Reviewer 2
The hypothesis is not clearly stated, and it should be set out and elaborated. Also, authors should point out aims in more details.
Thanks Reviewer 2. We have included this information in Lines 78-81.
Line 47: there is bracket that is either unnecessary or it should be closed
Thanks. This has been corrected.
It is unusual to use letters to describe statistical difference in t-test, are the authors using letters to be easier to read the statistics and to distinguish differences between each variety in 2017 and 2018?
Reviewer 2 is right. Letters have been used to make easier the understanding of the statistical analysis performed. Besides, letters were added to distinguish differences between years for each variety.
Why the authors calculate mean values for each parameter for both year and for all varieties in single year? Please explain since I don`t see the point of that.
Most of the parameters analyzed differed significantly according to the year, probably due to the effect of high temperatures in 2017. In some cases, the variety showed also a significant influence. On the other hand, differences between years were found in some cases depending on variety and/or the length of the cycle of the tested quinoa varieties. For this reason, results show mean values for each parameter, as they will indicate differences inherent to the year and/or the variety analyzed. As a brief example of how means can contribute to a better understanding of the results: crude protein (CP) (Table 1) was considerably higher in 2017. Because of the higher straw yield of 2018, the lower straw CP achieved in 2018 could be a consequence of a dilution effect. However, in 2017 the CP was also higher in the medium-long cycle varieties, which also achieved higher yields.
Line 290: reference 62 should be corrected
Thanks. This has been corrected
Methods for analysis and measurements should be described more and references should be added.
Thanks. References are added. If any method should be further explained, please, let us know.
Reviewer 3 Report
This is a very nice article reporting the response of high temperature in the yield and nutrition composition of quinoa straw.
Here are some comments:
Line 12: Why the abbreviation of (Climate Smart Agriculture). You used just once in all the document, actually just in the abstract.
Line 15. Change annuities per “year”
Line 21: Straw yield
Line 24: why all in capital letters? Climate Smart Agriculture is not named in all the document.
Line 36: change the symbol of degrees.
Line 47: You open parentheses but you never close it).
Line 57: Change high protein content. You can say high quality protein.
Line 68: erase or : and add Portugal (quinoa portuguesa Quem somos | QPT Quinoa Portuguesa)
Line 80: Erase point 2.2. Figures, Tables and schemes.
Line 102: Erase ; and replace by and
Line 103: Seed yield is not reported, why you mentioned here that HI it was not correlated with seed yield?
Line 140: Why just in the text mentioned the average of N and Mg, why not to mention P, K and Ca too?
Line 141-142. Clarify the sentence, looks confused.
Line 227: Again Seed yield is not reported, why you mentioned here that HI it was not correlated with seed yield? Seems, you reported seed yield in your other publication (36) with the same experiment.
Line 273: It will be good to present an example of one or two minerals with values.
Line 280: What is it is AFGC? American Forage and Grassland Council?? You should add the abbreviation in line 275.
Line 285: Winter must be with lowercase.
Line 295: In your previous work (36), seems the variety Marisma either in high temperature produce good production almost 2 t/ha. Have you have experience of this variety in commercial fields? It is still produce between 2-4 t/ha??
Line 296: One point in the future is to study the palatability in animals in quinoa straw, for experience most animals don’t like quinoa straw some varieties produce very hard and thick straws.
Line 321: Where is the origin of those varieties? Pasto, and Jessi are from Radicle (Netherlands) but could you mention about Roja, Duquesa and Marisma?
Line 330-331. You mean here samples = straw??
Line 358: You did not reported grain yield, but for the HI you need it. If you don’t report it will be good to mentioned that was used the grain yield from previous work (36).
Author Response
Many thanks Reviewer 3. We truly appreciate your comments that will greatly help to improve our manuscript.
Changes in the manuscript have been highlighted by yellow color. The overall language of the manuscript has been revised, and changes have been highlighted by text in a red colour font.
Here are some comments:
Line 12: Why the abbreviation of (Climate Smart Agriculture). You used just once in all the document, actually just in the abstract.
Reviewer 3 is right. This abbreviation has been deleted.
Line 15. Change annuities per “year”
Thanks. This term was incorrectly used, and we have changed this as recommended.
Line 21: Straw yield
Thanks. We have corrected this.
Line 24: why all in capital letters? Climate Smart Agriculture is not named in all the document.
Reviewer 3 is right. The term is used in a general way. This has been corrected (Line 12)
Line 36: change the symbol of degrees.
Thanks. This has been changed
Line 47: You open parentheses but you never close it).
Thanks. This has been corrected
Line 57: Change high protein content. You can say high-quality protein.
Thanks. This has been changed
Line 68: erase or : and add Portugal (quinoa portuguesa Quem somos | QPT Quinoa Portuguesa)
Done
Line 80: Erase point 2.2. Figures, Tables and schemes.
Done
Line 102: Erase ; and replace by and
Done
Line 103: Seed yield is not reported, why you mentioned here that HI it was not correlated with seed yield?
Thanks Reviewer 3 for raising this point. You are right, seed yield was reported in previous work. Thus, the sentence has been changed and seed yield is not considered.
Line 140: Why just in the text mentioned the average of N and Mg, why not to mention P, K and Ca too?
Thanks, and sorry. It was a bit confusing. We mentioned N and Mg exclusively because they were the two only nutrients that did not show a variety-related effect. We believe that now, changing a bit the sentence, has helped to clarify the information (Please, check Line 143).
Line 141-142. Clarify the sentence, looks confused.
Thanks. The sentence has been changed (Please, check Lines 144-146).
Line 227: Again Seed yield is not reported, why you mentioned here that HI it was not correlated with seed yield? Seems, you reported seed yield in your other publication (36) with the same experiment.
Reviewer 3 is right. The sentence has been changed and seed yield is not now considered
Line 273: It will be good to present an example of one or two minerals with values.
Indeed, this is a very good point. We have included this information in Lines 277-279 as suggested.
Line 280: What is it is AFGC? American Forage and Grassland Council?? You should add the abbreviation in line 275.
Sorry for not indicating this in the previous version. AFGC is indeed referred to the American Forage and Grassland Council. The abbreviation has been added accordingly. Thank you
Line 285: Winter must be with lowercase.
Thanks. This has been changed accordingly.
Line 295: In your previous work (36), seems the variety Marisma either in high temperature produce good production almost 2 t/ha. Have you have experience of this variety in commercial fields? It is still produce between 2-4 t/ha??
Thanks for raising this point. We have little experience in commercial fields, but we are closely working in collaboration with the Spanish company Algosur, as mentioned in the Acknowledgments section. This variety has been tested in commercial fields and is able to yield 2-4-t/ha
Line 296: One point in the future is to study the palatability in animals in quinoa straw, for experience most animals don’t like quinoa straw some varieties produce very hard and thick straws.
Again, Reviewer 3 is pointing to a very important aspect when considering the use of quinoa straws for animal feeding. Thank you for your interesting comment. In this study the planting density was high, being the stems relatively thin. On the other hand, we have observed really thick stems when the sowing date is delayed to spring or using long cycles varieties. We believe that animal feeding studies should be performed to analyze these aspects.
Line 321: Where is the origin of those varieties? Pasto, and Jessi are from Radicle (Netherlands) but could you mention about Roja, Duquesa and Marisma?
Thanks, and yes, Pasto and Jessie were coming from Wageningen. Roja, Duquesa and Marisma were quinoa varieties given by the company Algosur, as stated in the Acknowledgment section.
Line 330-331. You mean here samples = straw??
Yes. The sentence has been modified to clarify this aspect (Please, check Lines 336-337). “Samples” have been changed by “straw samples”.
Line 358: You did not reported grain yield, but for the HI you need it. If you don’t report it will be good to mentioned that was used the grain yield from previous work (36).
Reviewer 3 is right. We agree with this point. An explanation has been included (Please, check Lines 365-366).
Round 2
Reviewer 1 Report
The manuscript is improved.
However, I should highlight the recommendation to increase their number of replicates to increase the precision of the estimate of the mean and, thus, detect significant differences with a lower variation, for example, the 15% increase in 2017 and the 52% in 2018 in crude protein that in this work are not detected as significant.